# Identifying and Benchmarking Natural Out-of-Context Prediction Problems

**David Madras**
University of Toronto
Vector Institute
madras@cs.toronto.edu

**Richard Zemel**
University of Toronto
Vector Institute
Columbia University
zemel@cs.toronto.edu

## Abstract

Deep learning systems frequently fail at out-of-context (OOC) prediction, the problem of making reliable predictions on uncommon or unusual inputs or subgroups of the training distribution. To this end, a number of benchmarks for measuring OOC performance have been recently introduced. In this work, we introduce a framework unifying the literature on OOC performance measurement, and demonstrate how rich auxiliary information can be leveraged to identify candidate sets of OOC examples in existing datasets. We present NOOCH: a suite of naturally-occurring "challenge sets", and show how varying notions of context can be used to probe specific OOC failure modes. Experimentally, we explore the tradeoffs between various learning approaches on these challenge sets and demonstrate how the choices made in designing OOC benchmarks can yield varying conclusions.

## 1   Introduction

People often find context useful for prediction, both for improving accuracy and processing efficiency [10]. However, deep learning systems frequently over-rely on context cues [18, 19, 38], which can lead to poor performance on out-of-context (OOC) examples, when contextual information is misleading. By OOC examples, we mean inputs which are uncommon or unusual with respect to the training distribution; these can be thought of as sampled from under-represented subgroups, or low (non-zero) density regions, of the training distribution. In safety-critical situations, this can be problematic; as such, it is important to have reliable methods for measuring how well a model can perform OOC. Furthermore, given the rise of larger models and datasets [31], there is a need for scalable approaches to OOC evaluation — even if manual evaluation of "corner cases" by domain experts may (always) be the gold standard.

A key pre-requisite task to *evaluating* OOC performance is *identifying* which examples should be considered OOC. This identification task is a challenging one in and of itself: in a natural image, "context" can be varied, complex and high-dimensional [6, 47, 59, 63]. Therefore, any evaluation method intending to measure a model's OOC performance must (implicitly) select a specific notion of "OOC performance". Indeed, since deep learning yields underspecified models [14], it is plausible that different choices may yield different measurements. Common approaches include generating semi-synthetic data to simulate the effect of a shift in some salient feature [32, 54, 63], or using some auxiliary information to guide choices about what a reasonable OOC set should be [28, 33].

In this work, we develop a conceptual framework for identifying sets of OOC examples in existing datasets. We show how our framework unifies and generalizes prior literature on OOC performance measurement, and allows us to utilize more complex, structured annotated data to define various notions of "OOC performance". We demonstrate this framework's effectiveness through uncovering two OOC "challenge sets" [30] within an existing benchmark, each corresponding to differing notions

35th Conference on Neural Information Processing Systems (NeurIPS 2021).

of context. We show how our framework enables scalable and targeted measurement of models' OOC performance through clarifying the relationship between the concept of "OOC performance" and its implementation, allowing for clearer insight on current approaches as well as opportunities for improvement. Our contributions are as follows:

- We present NOOCH (Naturally-Occurring Out-of-context Challenge sets), a suite of "challenge sets" for evaluating performance on naturally-arising OOC problems, available at `https://github.com/dmadras/nooch`;
- We develop a conceptual framework for automatically identifying OOC challenge sets from existing data by leveraging known underlying structure;
- We contrast two instantiations of this framework using two notions of "context", defining concepts of "hard positives" and "hard negatives" in the OOC setting; and
- We quantitatively analyze the performance of several methods from the robust learning literature on these challenge sets, exploring the tradeoffs inherent in different approaches to OOC performance measurement; and qualitatively demonstrate how rich notions of context can yield rich investigation of OOC errors.

## 2 Measuring "Out-of-Context Performance"

Intuitively, a model which has good "OOC performance" should be able to maintain good performance under unusual or perturbed contextual conditions. We distinguish this from the out-of-distribution (OOD) problem [27], which is usually concerned with inputs from a different domain than the training set. Rather, the OOC prediction problem is more similar to subgroup robustness or distributional shift, where a model must perform well on uncommon input regions at training time. However, even after drawing this distinction, the notion is ill-defined: "context" may refer to concepts as varied as object relationships [59], image backgrounds [63], experimental settings [47], or world models [6]. Furthermore, even fixing a notion of context, the criterion for what should make something "out-of-context" (OOC) is still unclear. For instance, Peters et al. [47] focus on previously unobserved contexts (i.e. environments), whereas Xiao et al. [63] are concerned with unusual contexts given the class of interest (i.e. perturbations to image background).

Clearly, defining a benchmark to measure a method's OOC performance requires a number of design choices, which has enabled a recent proliferation of OOC benchmarks. We note in particular, one of the key choices is around the usage of *auxiliary information*. Across the literature on OOC performance measurement, there are a plethora of approaches to defining OOC criteria using some type of auxiliary information $C$. For the purposes of algorithm designers, $C$ may be assumed to be available at training, validation, and/or test time, or not at all — however, at the the time of benchmark design, it is available on a sufficiently large portion of the collected dataset to guide the designers' choices about what a suitable OOC criterion should be. Examining the current literature on measuring OOC performance, we identify the following as a unifying framework:

1. Identify some existing auxiliary information $C$, a variable which takes some value on many (or all) examples and specifies some underlying structure in the data.
2. Select a notion of "OOC" (e.g. "images with misleading backgrounds are OOC", "examples from unfamiliar time periods are OOC") and define an "OOC criterion" by choosing a binary function $\phi$ of $C$.
3. Restrict the test set to those examples where $\phi = 1$. Optionally, also restrict the training set to those examples where $\phi = 0$.

We show in Table 1 how a range of prior literature leverages auxiliary information to define OOC criteria. We can think of $C$ as providing benchmark designers with some type of "inductive bias" around what should be considered OOC for a given benchmark. The above framework implies that there is a diversity of OOC criteria which can be defined over any dataset, and this class is as broad as the class of functions $\phi$ which can be defined over the available auxiliary information. In the rest of the paper, we take advantage of the flexibility of this framework to give two examples of such an approach. We show that by leveraging more complex annotated structure, we can create multiple OOC benchmarks from an existing dataset using multiple criteria for what should be considered "OOC". We trace out how the choices made in designing these criteria correspond to different notions of context, and demonstrate experimentally that these yield varying measurements of OOC performance.

| Dataset | Auxiliary Information $C$ | OOC function $\phi$ |
|---|---|---|
| Waterbirds [54] | 1 if background is water (binary) | $C \neq Y$ |
| iWildCam2020-Wilds[33] | camera trap ID (categorical) | $C \notin \{1 \ldots 245\}$ |
| FMoW-Wilds [33] | time stamp (ordinal) | $C_{time} \geq 2013$ |
| Imagenet-A [28] | max NLL of ensemble (continuous) | $C \geq -\log(0.15)$ |
| Breeds [55] | subclass (categorical) | $C \in$ target subset |

Table 1: Examples of OOC benchmarks from the literature under our framework. The right-most column lists the condition under which $\phi = 1$. The OOC function $\phi$ in [28] has several additional filtering steps, some heavily manual.

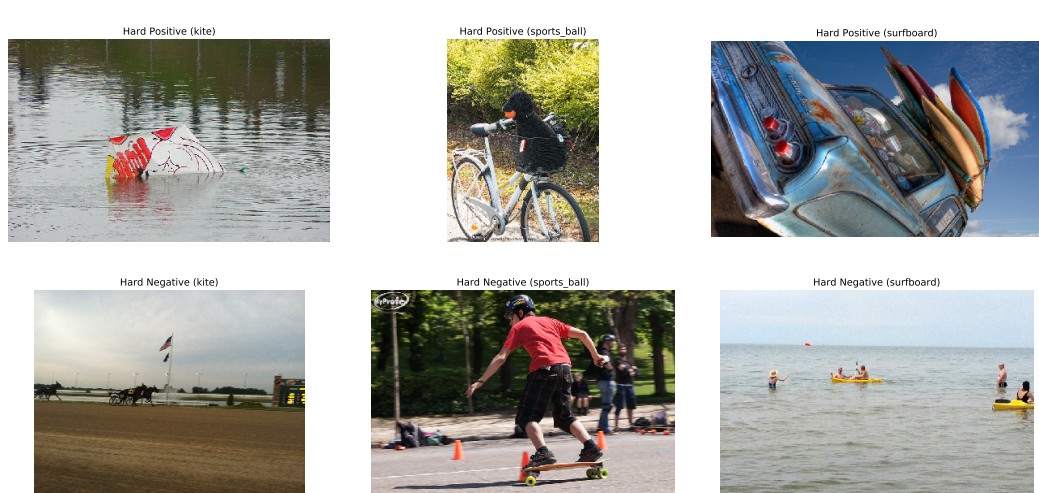

Figure 1: Using the co-occurrence/extractibility (CE) criterion, examples of (0.05, 0.1)-hard positives (top row) and negatives (bottom row) for the classes (L to R): `kite`, `sports_ball`, `surfboard`.

## 3 Finding Naturally-Occurring OOC Problems

We now demonstrate concretely how rich auxiliary information can be used to study the way that context shifts arise naturally within an existing computer vision benchmark, and provide two criteria for OOC performance that can be computed from these annotations. Throughout, we consider the binary prediction task of determining object presence, a problem where relationships between various objects naturally provide helpful context — given an image $X$, is an object of class $Y$ present or not?

**Background: COCO and COCO-Stuff.** The Microsoft Common Objects in COntext dataset (COCO) [36] is a computer vision dataset consisting of images of natural scenes. Each image is annotated with instance labels and segmentations for every "thing" in the image, as well as several captions describing the content of the scene. Images usually contain multiple items and as such usually have multiple labels. However, for the purposes of investigating OOC prediction, many relevant objects are not labelled in COCO; for instance, background objects such as "sky" or "grass" are not COCO classes. Fortunately, the COCO-Stuff dataset [7] provides labels and segmentations for all of the "stuff" in the images from COCO; a thing is an object with a specified size and shape, whereas stuff has no "defined spatial extent" [16]. Having both thing and stuff labels is essential for understanding model behaviour on OOC examples, since it is exactly these "stuff" classes which often (but not always) provide important context cues. Taken together, the thing and stuff annotations yield a rich sandbox for queries about the role of context in prediction. For our purposes, COCO-Stuff contains 171 binary tasks for determining object presence (81 thing classes and 90 stuff classes).

### 3.1 Automatically Identifying OOC Examples: Hard Positives and Negatives

We develop two contrasting notions of "OOC": 1. the presence/absence of frequently co-occurring, easily extractible objects; and 2. an unusual "gist" of a scene. We define these notions below, presenting two criteria for identifying naturally-occurring OOC prediction problems within the

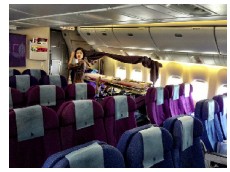 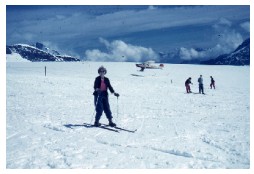 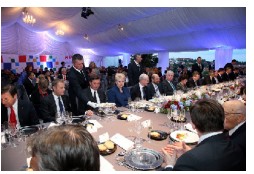 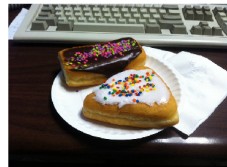

(a) hard positive (CE)  (b) hard positive (Gist)  (c) hard negative (CE)  (d) hard negative (Gist)

Figure 2: To contrast the CE and Gist criteria, we show samples from the `airplane` (L) and `bowl` (R) tasks.

existing COCO(-Stuff) dataset, and discussing how we can use annotations as proxies to define an OOC indicator $\phi$. We identify two types of OOC examples: *hard positives*, where the class is present despite an unusual context, and *hard negatives*, where the class is *not* present, despite a usual context.

### 3.1.1 Defining Context Using Co-Occurrences and Extractibility

For an object class $Y$, context cues often come in the form of another object class $C$ that has two properties (cf. [38]). First, $C$ and $Y$ co-occur frequently [5]. Second, $C$ is more *extractible* than $Y$ — easier to detect. If $C$ were *less* extractible than $Y$ it would not be a useful cue for detecting $Y$, as a model could detect $Y$ directly.

We can utilize these properties to create candidate context cues $C$ for a class of interest $Y$. Given segmentations, we can use an object's size within an image as a proxy for extractibility (larger objects tend to be more extractible). Let the $Area$ operator take the sum of the areas of all segmentations of instances of that object, or return 0 if the object is not present. Then, to estimate from the training set how important of a context variable $C$ is, we can compute $A(C, Y) = \mathbb{E}[Area(C) - Area(Y)|Y = 1]$. When $A(C, Y)$ is larger, this means that when $Y$ is present, $C$ is usually also present, and on average, takes up more of the image than $Y$ does. When $A(C, Y) > \alpha$, we say that $C$ is an $\alpha$-strong context cue for $Y$ (or just $\alpha$-context for brevity). We find that many of the contexts identified using this method for large enough $\alpha$ are intuitive. Some examples of (label, 0.05-context) pairs are: (`car`, `road`), (`bowl`, `dining_table`), (`cow`, `grass`).

Using this notion of context, we can then define hard positive and hard negative examples. We make the simplifying *noisy-or* assumption: that each context cue provides evidence for $Y$, so that the presence of any cue supports $Y$ being present, while the absence of all provides evidence against $Y$'s presence. Given some image, we can define an $(\alpha, \beta)$-hard positive or negative. If $Y = 1$, and for all $\alpha$-context cues $C$, we have $Area(C) < \beta$ in this image (and there is at least one $\alpha$-context cue), then the example is an $(\alpha, \beta)$-hard positive. Alternatively, if $Y = 0$, and there exists some $\alpha$-context variable $C$ such that $Area(C) > \beta$, then the example is an $(\alpha, \beta)$-hard negative. We will call this method the co-occurrence/extractibility (CE) criterion (see Fig 1 for examples). Throughout, we use $\alpha = 0.05, \beta = 1$ unless otherwise noted; these parameters were chosen since they approximately equalize $P[(X, Y)$ is $(\alpha, \beta)$-hard $|Y = y]$ across $y = 0, 1$. See Appendix B for more details.

### 3.1.2 Defining Context Using Gist

We now turn to a broader notion of context, that of the "gist" of a scene [59], or its overall semantic content. This is something that humans can recognize easily [35], but goes beyond object frequency and extractibility. When an object is present in a scene whose gist is very different from the scenes the object was present in at training time, this may make prediction difficult.

We describe our method for estimating gist shift, which we call the gist criterion. We use caption annotation data in COCO (each image has 5 caption sentences), making the assumption that information in a caption captures the gist of a scene. We then take an SBERT embedding [48] of each image caption for an image, and average these embeddings to get a single embedding for that image. Then, for a given image and some target label $Y$, we find at the cosine similarity between that image's embedding and the average embedding across all training images with $Y = 1$. If this similarity is below some threshold $\tau$, and $Y = 1$ for the test image, it is a $\tau$-hard positive; if this similarity is above $\tau$, and $Y = 0$ for the test image, it is a $\tau$-hard negative. Note, we do not look at distance to $Y = 0$ examples; we assume that captions for $Y = 0$ images may have little in common, whereas the mean caption for $Y = 1$ is a prototypical description of $Y$ "in context". Throughout, we set

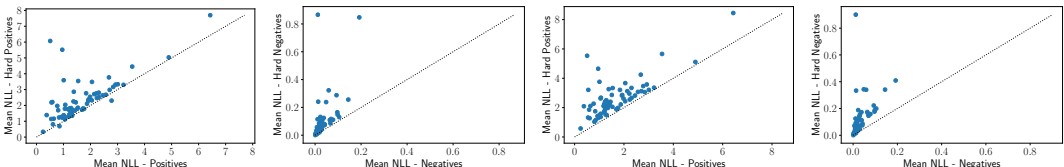

Figure 3: We find that hard positives/negatives induce higher average loss for both criteria. Each point is a task in COCO-Stuff: the x- and y-axis values show the average loss achieved by an ERM model on all positives (negatives) and the average loss on hard positives (negatives). From L to R: CE criterion (positives), CE criterion (negatives), gist criterion (positives), gist criterion (negatives). The diagonal line represents where the hard example losses match marginal losses.

the threshold $\tau$ for each task so that the number of hard positives and negatives is the same as the CE criterion for that task, to facilitate comparisons. See Figure 2 for examples of hard positive and negatives chosen by this criterion in comparison to the CE criterion, and Appendix B for more details.

### 3.2 NOOCH-CE and NOOCH-Gist: Selecting Challenge Sets

We train binary classifiers to minimize average NLL on each of the 171 classes in COCO-Stuff. We find that for nearly all tasks, the hard positives and negatives defined by our methods incur higher average loss than positive and negative examples respectively (Fig. 3), for both the CE and Gist criteria. This provides some evidence that our criteria are, in fact, identifying examples which are more difficult to classify correctly.

To select candidate OOC tasks for our challenge sets, we select the 12 tasks with the largest difference between average NLL on hard examples (by the CE criterion) and average NLL on all examples. We call these tasks the NOOCH (Naturally-Occurring Out-of-context Challenge) suite (Table 5). We then identify two groups of challenge sets: NOOCH-CE, which consists of the hard positive and negative examples on each of the 12 tasks in NOOCH as identified by the CE criterion; and NOOCH-Gist, which is the analogous set for the gist criterion. These tasks are: `car`, `bowl`, `boat`, `fire-hydrant`, `airplane`, `cow`, `backpack`, `cup`, `surfboard`, `tie`, `sports-ball`, `kite`. This gives us 24 total challenge sets on which to evaluate an ML model's OOC performance.

**Contrasting the Two Criteria (Fig. 2).**   The left two images in Fig. 2 are hard positives on the `airplane` task. On the far left, we see the inside of an airplane: this is selected as a hard positive by the CE criteria because there is no sky visible. On the second left, the sky is visible but the overall scene is unusual; this is selected as a hard positive by the Gist criteria. The right two images are hard negatives on the `bowl` task. On the second right, we see a large dinner with many plates: this is selected as a hard negative by the CE criteria because there is a prominent dining table but no bowls. On the far right, we see a paper plate on a desk (not a dining table): this is selected as a hard negative by the Gist criteria since there is no bowl but it is similar to images where you might expect a bowl.

## 4   Evaluating Robustness Approaches on NOOCH

We now turn to evaluating various approaches on the NOOCH benchmarks. We focus on four categories which provide a useful contrast between different approaches to OOC prediction: expected risk minimization, label-based adjustments, environment-based methods, and adaptive methods.

**Notation.** We are given a dataset $\{x_i, y_i\}_{i=1}^n$ with inputs and target labels respectively, and possibly some side information $\{c_i\}_{i=1}^n$ as well (e.g. environment variables). If side information is available, we assume it is available at training time but not necessarily test time. We aim to learn a function $f : \mathbb{X} \longrightarrow \mathbb{Y}$. We assume $\ell$ to be the the example-wise cross-entropy loss function.

**Expected Risk Minimization.**   Expected risk minimization (ERM) is the standard paradigm for training ML models; Gulrajani and Lopez-Paz [20] show it can be difficult to beat on domain generalization problems. In ERM, we minimize the mean loss $\ell$ on the training set: $\mathcal{L}_{ERM}(f) = \frac{1}{n}\sum_{i=1}^n \ell(f(x_i), y_i)$.

**Label-Based Adjustments.** Let $w_i = P(Y = y_i)$ in the training set. For fairer comparison with environment-based methods below, we add a tuning parameter $\alpha$ to control the degree of adjustment. $\alpha = 1$ represents the standard versions of these loss functions, but we found other values $\alpha \in [0, 2]$ to be useful. We weight the loss for each example proportional to $(\frac{1}{w_i})^\alpha$ (label reweighting - **Reweight**), or sample each point with probability proportional to $(\frac{1}{w_i})^\alpha$ (label undersampling - **US**).

## 4.1 Environment-Based Methods

One common setting is where the auxiliary information $c_i$ for each example is a categorical variable, representing an *environment*. This type of grouping procedure can be used separately from its original causal context [47]; we consider it to represent some informative partition of our data. Here, $c \in \{1 \ldots C\}$ is an integer, $n_c$ is the number of examples in environment $c$, and the average loss for an environment is: $\ell_c(f) = \frac{1}{n_c} \sum_{i=1}^n \ell(f(x_i), y_i) \mathbb{1}\{c_i = c\}$.

**Group DRO.** Group Distributionally Robust Optimization (GDRO) [54] aims to minimize the loss on the worst of $C$ partitions. The group adjustment with hyperparameter $K$ ensures greater focus on smaller groups: $\mathcal{L}_{GDRO}(f) = \max_{c \in \{1 \ldots C\}} \left\{ \ell_c(f) + \frac{K}{\sqrt{n_c}} \right\}$.

**IRM.** Invariant risk minimization (IRM) [2], uses a gradient penalty on the output of $f$, with $w$ a constant multiplier on the output of $f$, and hyperparameter $\lambda$. The intuition is somewhat involved, but the motivation is to learn a representation such that the same predictive classifier is optimal across environments: $\mathcal{L}_{IRM}(f) = \sum_{c=1}^C \ell_c(f) + \lambda \|\nabla_{w|w=1} \ell_c(w \cdot f)\|$.

**Environment Reweighting and Undersampling.** These are equivalent to label reweighting/undersampling above, but with $w_i = P(C = c_i)$ (**Reweight-Envs**, **US-Envs**).

## 4.2 Adaptive Methods

We also consider loss functions which, rather than using side information to specify which examples are OOC, focus dynamically on the hardest examples at each step. In conditional variance-at-risk optimization (**CVaR**) [49], we aim to minimize the loss over a worst-case distribution over training examples. For some $p \in (0, 1)$, CVaR($p$) is defined as the average loss of the $p$-percent worst-loss examples. In **focal** loss [37], we dynamically upweight high loss examples using a parameter $\gamma \geq 0$. With binary $y$, let $q(x, y) = yf(x) + (1 - y)(1 - f(x))$. Then, at $\gamma = 0$ focal loss reduces to cross-entropy; as $\gamma$ increases, it focuses more on the examples which have higher loss already: $\mathcal{L}_{Focal}(f) = \frac{1}{n} \sum_{i=1}^n -(1 - q(x_i, y))^\gamma \log(q(x_i, y))$.

# 5 Related Work

A range of datasets have been proposed for the purposes of benchmarking OOC prediction. Several focus on realistic OOC prediction: Koh et al. [33] contains problems from across a range of applications; Hendrycks et al. [28] select examples which perform worst on an ensemble of models; Barbu et al. [4] focuses on object recognition and shows objects varied by a range of attributes; Choi et al. [8] isolates 26 images of objects in unusual contexts from a larger dataset. Our work functions well as a complement to any of these datasets; we believe it is novel due to our ideas for scalably identifying challenge sets from annotated data, as well as its delineation of hard positives and hard negatives. The notion of "challenge sets" [30], "stress tests" [41], or "contrast sets" [17] from the NLP literature is an inspiration for our work as well. A range of primarily semi-synthetic datasets include those that center around: image corruption [26, 39]; object hierarchy [55]; synthetic shifts in background [54, 63]; color [32]; a group attribute [1]; or purely synthetic data [3].

Several works have discussed explicit examples where deep models failed to perform OOC prediction in practice. Oakden-Rayner et al. [44] and Winkler et al. [61] discuss the risk of this occurring in the medical domain, and Shetty et al. [57] in the autonomous driving domain. Other works have detailed the challenge of OOC prediction for deep models, using frames of "shortcuts" [19], "simplicity" [56], "extractibility" [38], texture biases in CNNs [18], or the challenge of out-of-place objects [50].

| Task | ERM | CVaR | Focal | Reweight | US | Reweight (Envs) | US (Envs) | GDRO | IRM |
|------|-----|------|-------|----------|-----|----------------|-----------|------|-----|
| car | 0.769 | 0.787 | 0.759 | 0.773 | 0.773 | **0.886** | 0.846 | **0.891** | 0.868 |
| bowl | 0.749 | 0.781 | 0.734 | 0.751 | 0.759 | **0.864** | 0.814 | **0.865** | 0.828 |
| boat | 0.869 | 0.888 | 0.823 | 0.866 | 0.877 | **0.954** | 0.923 | **0.945** | 0.925 |
| fire-hydrant | 0.913 | 0.933 | 0.908 | 0.927 | 0.913 | 0.933 | 0.920 | **0.946** | **0.942** |
| airplane | 0.986 | 0.984 | 0.983 | 0.986 | 0.985 | **0.991** | 0.986 | **0.991** | 0.986 |
| cow | 0.935 | 0.937 | 0.932 | 0.939 | 0.938 | **0.963** | 0.948 | **0.963** | 0.943 |
| backpack | 0.812 | 0.806 | 0.809 | 0.816 | 0.812 | 0.813 | 0.816 | **0.871** | 0.844 |
| cup | **0.870** | 0.863 | 0.867 | **0.876** | **0.873** | **0.879** | **0.875** | 0.825 | **0.869** |
| surfboard | 0.939 | 0.947 | 0.933 | 0.940 | 0.939 | **0.960** | 0.940 | **0.960** | **0.957** |
| tie | 0.742 | 0.752 | 0.728 | 0.760 | 0.763 | 0.756 | 0.761 | **0.806** | 0.775 |
| sports-ball | 0.867 | 0.869 | 0.871 | 0.869 | 0.868 | **0.911** | 0.890 | **0.911** | 0.894 |
| kite | 0.932 | 0.932 | 0.937 | 0.940 | 0.928 | 0.949 | 0.936 | **0.960** | **0.950** |
| Average | 0.865 | 0.873 | 0.857 | 0.870 | 0.869 | 0.905 | 0.888 | **0.911** | 0.899 |

Table 2: AUC on hard test examples for all 12 NOOCH-CE stress tests, after hyperparameter selection. Bold numbers have overlapping standard deviations with the highest observed mean's.

| Task | ERM | CVaR | Focal | Reweight | US | Reweight (Envs) | US (Envs) | GDRO | IRM |
|------|-----|------|-------|----------|-----|----------------|-----------|------|-----|
| car | 0.766 | 0.785 | 0.763 | 0.768 | 0.773 | 0.845 | 0.823 | **0.863** | 0.832 |
| bowl | 0.642 | **0.686** | 0.635 | 0.656 | 0.678 | **0.692** | **0.698** | **0.704** | **0.670** |
| boat | 0.826 | 0.855 | 0.771 | 0.823 | 0.823 | **0.897** | 0.868 | 0.884 | 0.872 |
| fire-hydrant | 0.840 | **0.865** | 0.822 | 0.849 | 0.841 | 0.834 | 0.836 | 0.823 | 0.845 |
| airplane | **0.977** | **0.979** | 0.975 | **0.978** | 0.976 | **0.978** | 0.977 | **0.981** | **0.977** |
| cow | 0.901 | **0.912** | 0.877 | 0.903 | 0.906 | 0.908 | **0.908** | **0.911** | 0.896 |
| backpack | 0.716 | 0.717 | 0.725 | 0.731 | **0.749** | 0.729 | 0.736 | **0.732** | **0.750** |
| cup | 0.733 | 0.738 | 0.727 | 0.735 | 0.742 | 0.759 | **0.771** | 0.742 | **0.755** |
| surfboard | 0.913 | **0.920** | 0.900 | 0.912 | 0.914 | 0.909 | 0.912 | 0.882 | 0.894 |
| tie | **0.822** | 0.816 | 0.813 | **0.829** | **0.824** | **0.829** | **0.831** | **0.835** | **0.840** |
| sports-ball | 0.830 | 0.832 | 0.828 | 0.830 | 0.822 | 0.866 | 0.859 | **0.880** | 0.855 |
| kite | **0.942** | **0.939** | **0.945** | **0.947** | 0.936 | **0.947** | **0.939** | **0.947** | **0.943** |
| Average | 0.826 | 0.837 | 0.815 | 0.830 | 0.832 | **0.849** | 0.847 | **0.849** | 0.844 |

Table 3: AUC on hard test examples for all 12 NOOCH-Gist stress tests, after hyperparameter selection. Bold numbers have overlapping standard deviations with the highest observed mean's.

| Metric | ERM | CVaR | Focal | Reweight | US | Reweight (Envs) | US (Envs) | GDRO | IRM |
|--------|-----|------|-------|----------|-----|----------------|-----------|------|-----|
| Worst-Group Error | 0.4 | 0.354 | 0.402 | 0.361 | 0.336 | 0.233 | 0.146 | **0.108** | 0.127 |
| Worst-Group NLL | 1.03 | 0.657 | 0.722 | 0.884 | 0.883 | 0.535 | 0.417 | **0.306** | **0.346** |
| AUC-Hard | 0.691 | 0.703 | 0.506 | 0.657 | 0.701 | 0.744 | 0.778 | **0.929** | 0.788 |

Table 4: AUC on "hard" test examples on the Waterbirds dataset. Hard examples are the union of two groups: land birds on water backgrounds and water birds on land backgrounds. Worst-group error or NLL takes the worse of the two; AUC-Hard calculates the AUC across the union.

A range of work outside deep learning considers the OOC prediction problem from a different direction, focusing on how to improve prediction by taking context into account [9, 25, 40, 65]. Other work looks at the idea of using a latent variable to represent scene gist [45, 62]. A number of newer methods not discussed elsewhere in the paper also aim to solve the OOC problem, including those that involve side information [29, 34, 58, 64], those that involve side information through causal underpinnings [24, 52], and those that ignore side information altogether [12, 13, 15].

## 6 Experiments

In this section, we compare and contrast the various measurements of OOC performance yielded by NOOCH, along with the semi-synthetic Waterbirds dataset [54]. For all experiments we use a ResNet-50 [23], finetuned from ImageNet-pretrained features [53]. See App. B and C for further experimental details. For the environment-based methods, we follow Sagawa et al. [54] and create 4 environments: 1 for each element of the cross-product of the label and its highest-$\alpha$ context class. Many of the robust baselines from Sec. 4 come with a hyperparameter which aims to trade off between average performance and OOC performance; we choose the hyperparameter which minimizes the maximum loss of hard positives and hard negatives on the validation set.

### 6.1 Quantitative Analysis

#### 6.1.1 Main Results: Different Criteria Yield Different Evaluations

In Tables 2 and 3, we show AUC (area under the ROC curve) on hard examples for each method we focus on and each of the 12 NOOCH tasks individually. We choose AUC as a metric since it is robust to label imbalance, and tasks in NOOCH contain 1-10% positive examples.

For comparison, we show in Table 4 an analogous table of results for Waterbirds [54], a semi-synthetic dataset which is generated by pasting images of birds on top of either either land or water backgrounds; the goal is to classify land birds from water birds. At training time, land birds are mostly shown on land (and water birds on water), but at test time we hope to perform well, regardless of background. The information regarding background is made available through environments: the four environments are land/land, land/water, etc. For Waterbirds, we show both AUC-Hard (our metric) and worst-group erorr/NLL (the metrics from Sagawa et al. [54]).

**Environments are More Useful on Simpler Benchmarks.** We note that the three benchmarks yield varying conclusions about the methods in question; in particular, the relative performance of the best environment-based methods vary greatly between the benchmarks. We find that environment-based methods perform better on the benchmarks where the context shift's structure (i.e. the form of $\phi$) is better specified by the environments. In Table 4, we see there is a very large gap in performance between the best environment-based methods (particularly GDRO) and the methods that do not use environments. For instance, GDRO and IRM improve over ERM by about 0.3 in worst-group error (the metric of choice in Sagawa et al. [54]); and GDRO improves over *all* other methods by about 0.14 in AUC on hard examples. This difference is an order of magnitude greater than observed on either NOOCH benchmark, suggesting that semi-synthetic datasets (such as Waterbirds), may overestimate the performance of current environment-based methods, and possibly GDRO in particular.

We also see this when comparing NOOCH-CE to NOOCH-Gist: the environment-based methods are also the ones whose performance falls off the most from NOOCH-CE to NOOCH-Gist. This could be because NOOCH-CE is a simpler benchmark, whose notion of OOC is more well specified by the given environments. The contrast between Tables 2, 3 and Table 4 documents the usefulness of having benchmarks for robustness across a range of complexity, and motivates the creation of benchmarks such as NOOCH.

**Gist Shift is Difficult.** We further note that performance on NOOCH-Gist is generally worse than on NOOCH-CE. This suggests that the more complex notion of context embodied by the gist yields a more difficult OOC task. In some ways, would expect models to find these more holistic shifts harder, as these gist shifts go well beyond "shortcuts" [19].

**Access to Auxiliary Information is More Important than Algorithm.** Overall, environment-based methods perform the best on all three benchmarks — we expect this to be the case, since these methods are given access to structure which is relevant to the OOC task at hand. This is mostly clearly indicated in the improvement between reweighting/undersampling methods when using environments rather than labels. In fact, we find that on the more complex NOOCH benchmarks, reweighting examples by environment performs similarly to more specialized methods such as GDRO/IRM, given an equivalent amount of hyperparameter tuning. This suggests that current environment-based algorithms have significant room to improve when it comes to more complex OOC benchmarks.

#### 6.1.2 Secondary Observations

In Figures 4, 5, and 6, we show AUC, mean negative log-likelihood (NLL), and expected calibration error (ECE) [21] respectively, averaged across the 12 tasks. We show results for classification error in App. C.5. For AUC and ECE, we display results for hard and easy examples separately (where an "easy" example is defined as one which is not "hard", i.e. not a hard positive or hard negative). For NLL, we further break out the results on hard examples into hard positives and hard negatives. For all, we show NOOCH-CE (L) and NOOCH-Gist (R).

**Tradeoffs Exist Between Harder and Easier Examples.** We note two tradeoffs across AUC, and NLL results: models that are better on hard examples tend to be worse on easy examples (and vice

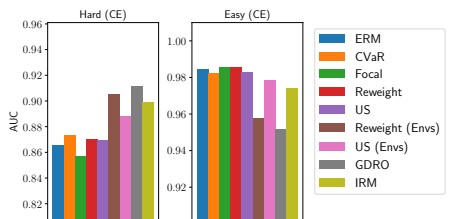
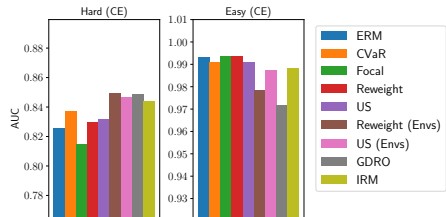

Figure 4: AUC (area under the ROC curve) achieved on hard and easy examples. Higher is better.

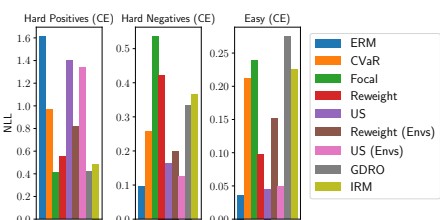
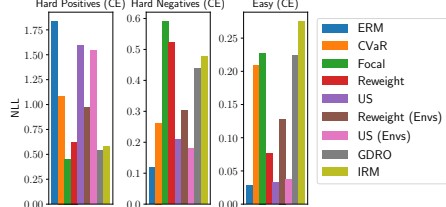

Figure 5: Negative log-likelihood (NLL) achieved on hard positive, hard negative, and easy examples.

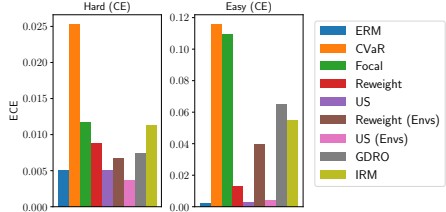
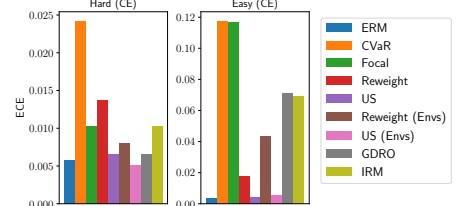

Figure 6: Expected Calibration Error (ECE) achieved on hard and easy examples. Lower is better.

versa), and models that are better on hard positives tend to be worse on hard negatives. As we expect, ERM is perfroms better on easy examples and hard negatives: since the dataset is imbalanced, negatives are the majority class and the hard negatives are the easier "hard" examples.

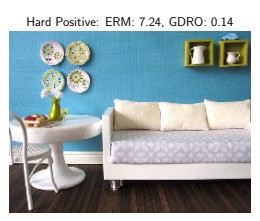
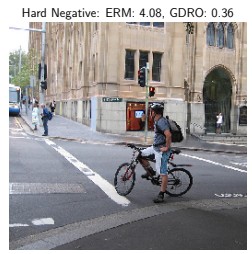

Figure 7: Test set examples where GDRO most improves over ERM (L) hard positives and (R) hard negatives from the NOOCH-CE `car` category. Titles show NLL for each method on that image.

**Adaptive Methods Find Different Tradeoffs.** We find that CVaR and focal loss find interesting tradeoffs between ERM and environment-based methods. CVaR performs comparably to ERM on both hard and easy examples by AUC ; and by ECE it is by far the worst of any method. However, CVaR's NLL on hard positives is between ERM and the environment-based methods'. Focal loss, on the other hand, performs similarly to ERM by AUC on both hard and easy examples, but it performs similarly to the environment-based methods on hard positives and negatives by NLL, providing a strong tradeoff between overall and OOC performance.

**Two Surprising Calibration Results.** We found that there was *not* the same tradeoff with calibration as there was with the other metrics. In particular, ERM and undersampling had the best calibration on *both* hard and easy examples, suggesting a path forward to avoid tradeoffs between OOC and average performance by thinking more about calibration. Secondly, we found the environment-based methods were not better calibrated, even on hard examples.

## 6.2 Qualitative Analysis

**Where Environment-based Learning Improves Over ERM.** In Fig. 7, we show the images with the largest gaps in NLL on the `car` task among hard positives and hard negatives (in NOOCH-CE). The hard positive and hard negative where GDRO most overperformed are both images we might expect, where the object label and context do not match: a living room with a tiny toy car on the shelf, and a normal street scene that happens to not have any cars. See App. D for analysis on where GDRO underperforms ERM.

**Contrasting NOOCH-CE and NOOCH-Gist.** In Fig. 8, we use examples of hard positives from the `cow` task to contrast the CE and gist criteria. On the left, we show a hard positive from NOOCH-CE, with cows standing on pavement: this is a hard positive in NOOCH-CE since there is no grass; GDRO and IRM outperform ERM on this image. However, it is *not* a hard positive in NOOCH-Gist, since cows are the central focus of the image. On the right, we show a hard positive from NOOCH-Gist, where the image focuses on a giraffe, but there are several white cows in the background standing on a grassy field. This is *not* a hard positive in NOOCH-CE, due to the large field, but it *is* a hard positive in NOOCH-Gist, since the giraffe is the focus of the image. ERM outperforms GDRO and IRM on this image — the environment-based objectives do not encourage as strong performance where the context (grass) and object (cow) align.

# 7 Discussion

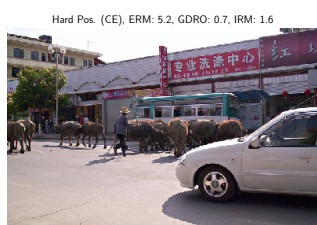
Hard Pos. (CE), ERM: 5.2, GDRO: 0.7, IRM: 1.6

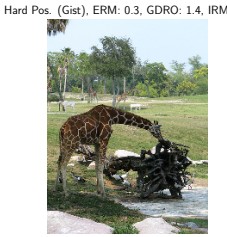
Hard Pos. (Gist), ERM: 0.3, GDRO: 1.4, IRM: 1.6

Figure 8: Examples of hard positives from NOOCH-CE (L) and NOOCH-Gist (R) for the `cow` task. Titles show NLL for several methods on that image.

**Limitations.** The idea of computationally identifying OOC examples is necessarily limited: by their nature, OOC examples are exceptions to rules, and there will always be OOC examples which can only be discovered through qualitative examination. Further, the task chosen for our benchmark was chosen for simplicity of analysis in a research context rather than real world significance — outputting segmentations or many-way classification may be more applicable to most applications. We hope that the impact of our work is to enable better evaluation of model performance on atypical or under-represented examples, both through usage of these challenge sets and ones inspired by the ideas presented here. However, the usage of benchmarks can have downsides where standardized performance metrics are prioritized over fundamental advances in modelling which are not captured in those metrics. In various applications of interest, results may need to be reported across a number of different metrics since this gives a clearer picture of model performance — AUC may not always be a (or the most) relevant metric.

**Conclusion & Looking Forward.** In this work, we study OOC evaluation as its own field, drawing attention to the range of evaluation schemes which can be used and the ways in which they may differ. We demonstrate that using auxiliary structure in the data can be a useful means of defining context, and that this structure can be used in rich ways to identify various faces of the OOC problem. Through this exploration, we find that methods which take advantage of auxiliary information may be more generously evaluated by OOC benchmarks which more are cleanly defined by that information.

In closing, we would like to highlight the idea systematizing OOC evaluation, whether this is through automatic discovery of OOC examples through annotations or some other means. This not only enables the scalable creation of challenge sets, but allows for faster generation and exploration of new hypotheses about the type of context shifts that models struggle on, possibly analogous to automatic test generation in the debugging literature [11, 42]. We hope that methods of this type can be applied elsewhere to better understand the challenges of OOC prediction.

## Acknowledgments and Disclosure of Funding

Thanks to Marc-Etienne Brunet, Eleni Triantafilliou and Elliot Creager for their helpful thoughts on the manuscript, as well as to four anonymous reviewers for useful feedback. David Madras was supported by an NSERC Alexander Graham Bell Canada Graduate Scholarship-Doctoral (CGS-D). Resources used in preparing this research were provided, in part, by the Province of Ontario, the Government of Canada through CIFAR, and companies sponsoring the Vector Institute (`www.vectorinstitute.ai/#partners`).

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
