# A  Other Methods

We detail a few other methods which we explored but do not include for analysis in Sec. 6. We found the methods we included in the paper had the best results overall, on the standard and hard examples.

- Logistic regression on pre-trained Imagenet features: We found this performed noticeably worse than ERM across the board, with a slight improvement in loss on hard positives (but worse than most other methods).
- ERM trained from scratch: We found this to perform slightly worse across the board than ERM from a pre-trained model.
- ERM with learning rate decay: We explored several learning rate decays and found, with tuning, minor improvements in overall performance from ERM. Due to the level of tuning required, we chose not to include these results and just used a constant learning rate.
- ERM with data augmentation: We explored using data augmentations such as random affine transformations, cropping, and horizontal flips, but found they did not improve performance in ERM.
- Adaptive parameters [46, 60]: We explored the possibility of learning affine transformations after each ResNet block. With tuning, we found this improved performance on hard examples over ERM but not close to the level of the other robust methods shown.
- Auxiliary prediction: We explored adding a second readout head to the final layer and predicting the highest $\alpha$-context variable as an auxiliary loss. We found this to perform nearly identically to ERM in all metrics, and occasionally slightly worse.

We include this list for completeness: this is not to say that no such approach can be useful for this problem, but at the time of writing we did not find compelling enough results from these methods to yield interesting or informative comparative analysis in Sec. 6.

# B  Data

## B.1  Licences

- The images in the COCO dataset [36] are provided under the Flickr terms of use.
- The annotations in the COCO dataset [36] are provided under a Creative Commons Attribution 4.0 License.
- The annotations in the COCO-Stuff dataset [7] are provided under a Creative Commons Attribution 4.0 License.

## B.2  Dataset Details

We use the COCO-Stuff dataset [7] (`https://github.com/nightrome/cocostuff`), which contains images and annotaitons from the COCO dataset [36], as well as adding additional annotations. Since COCO is a competitive benchmark, the test set is not publicly available, so we merge the provided training/validation sets and create our own new training/validation/test split using 70/10/20 percentages. This yields training/validation/test set sizes of 86302/12328/24657.

## B.3  Creation of the NOOCH Suite

When choosing the tasks, we filtered out a couple of tasks which seemed subjectively too similar to ones already in the benchmark. Specifically, we removed `sheep`, `handbag`, `bottle`, and `wine-glass` due to the fact that other similar tasks had higher average difference in average NLL on all examples and average NLL on hard examples (specifically, `cow`, `backpack`, `cup`, and `cup` respectively). We show in Table 5 how many hard positives, hard negatives, and positives are present in the test set for each task in NOOCH.

Fig 9 shows that, under the CE criterion, most of the 171 object classes in COCO-Stuff have context cues for $\alpha \geq 0$. Note that a 0-context cue is fairly strong: this means that on average, in images where $Y$ is present, $C$ takes up at least as much area as $Y$ does. This indicates that there may be a large

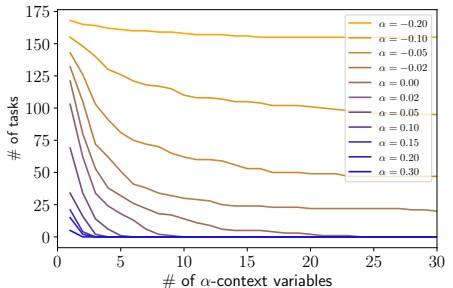

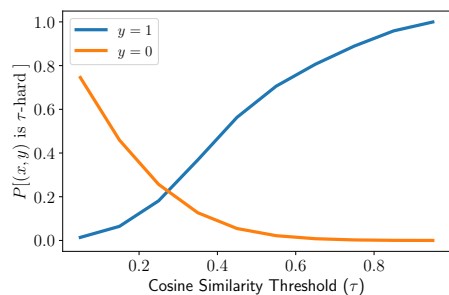

Figure 9: The number of tasks in the COCO-Stuff dataset which have context variables identifiable at each level of $\alpha$. There are 171 total tasks. For instance, this figure shows that there are around 70 tasks with at least one context cue at $\alpha \geq 0.05$, and around 30 tasks with at least five context cues at $\alpha \geq 0$.

Figure 10: Using the gist criterion, the number of $\tau$-hard positive/negative examples in the COCO-Stuff test set as a proportion of all positive/negative examples, where $\tau$ thresholds the cosine similarity to the average caption embedding of that category in the training set.

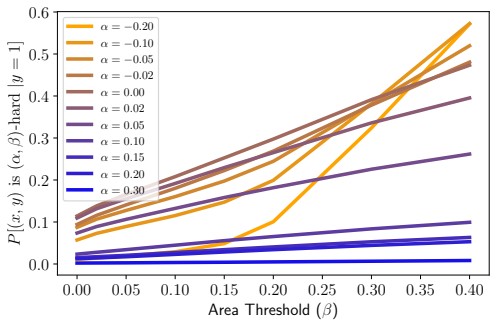

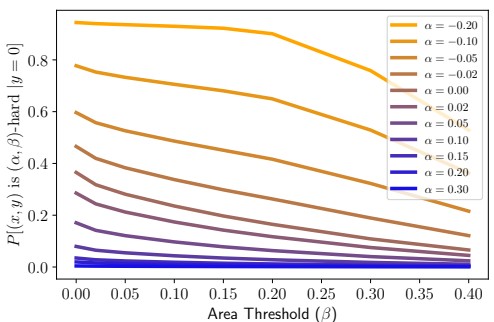

Figure 11: Across a range of $\alpha, \beta$, the percentage of hard positive/negative examples (L/R) in the test set which are $(\alpha, \beta)$-hard.

number of OOC examples hidden in many of these tasks. We find that the contexts returned using this method for large enough $\alpha$ are intuitive. Some examples of (label, 0.05-context) pairs are: (car, road), (bowl, dining_table), (cow, grass). Additionally, many tasks have more than one context variable: at $\alpha = 0$ about 50 tasks have 3, and at $\alpha = 0.05$ about 15 tasks have 3. Some examples of labels with multiple 0.05-context variables are: (surfboard, [sea, sky-other]), (tennis-racket, [person, playing-field]), (traffic-light, [building-other, road, sky-other, tree]). All of these groupings are intuitive, providing some evidence that this method of isolating context variables may be a useful one.

In Fig. 10 we see, across all values of $\tau$, how many $\tau$-hard examples there are in the test set by the gist criterion, across all 171 tasks in COCO-Stuff. As expected, the number of hard positives increases with $\tau$, and the number of hard negatives decreased.

Fig 11 shows that hard positive/negative examples are numerous in the data. For instance, we see that at the (0.05, 0.1) level, about 10% of all possible (object, category) pairs are hard positives, and a similar number for hard negatives.

Fig. 12 shows the average loss incurred by $(\alpha, \beta)$-hard positive and negative examples under the CE criterion as $\alpha, \beta$ vary.

| Task | # Hard + | # Hard - | # + |
|---|---|---|---|
| car | 1539 | 949 | 2585 |
| bowl | 944 | 1084 | 1463 |
| boat | 361 | 756 | 649 |
| fire-hydrant | 189 | 1577 | 332 |
| airplane | 258 | 3622 | 635 |
| cow | 194 | 2360 | 445 |
| backpack | 607 | 6413 | 1183 |
| cup | 743 | 6852 | 1926 |
| surfboard | 147 | 4011 | 750 |
| tie | 136 | 6347 | 778 |
| sports-ball | 151 | 6574 | 860 |
| kite | 55 | 6725 | 451 |

Table 5: Counts of hard positive, hard negative & positive examples in the test set per NOOCH task.

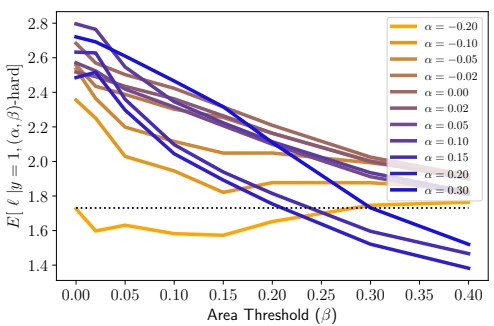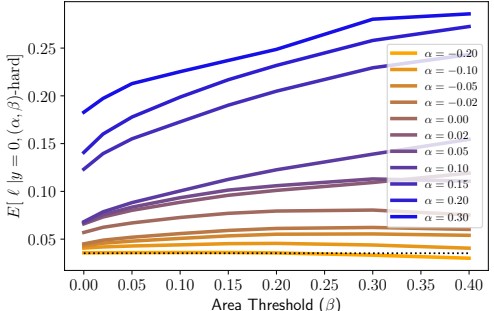

Figure 12: Across a range of $\alpha, \beta$, the average test set loss of positive/negative examples (L/R) in the test set which are $(\alpha, \beta)$-hard.

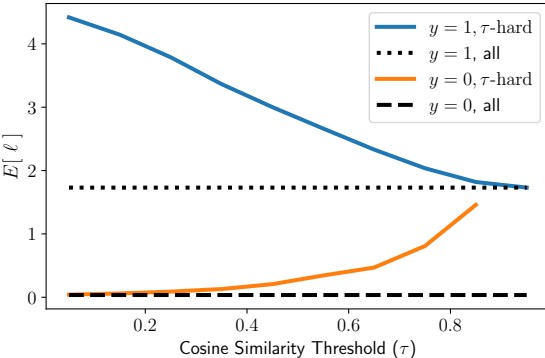

Figure 13: Caption

Fig. 13 shows the average loss incurred by $(\tau)$-hard positive and negative examples under the gist criterion as $\tau$ varies.

In the paper, we choose $\alpha, \beta = 0.05, 0.1$ for the CE criterion, and then set $\tau$ in each task such that the number of hard positive and negative examples in each task is the same. Tables 6 and 7 show these values of $\tau$ for each for the 12 NOOCH tasks.

| Task | Hard Positives | Hard Negatives |
|---|---|---|
| car | 0.48 | 0.48 |
| bowl | 0.64 | 0.57 |
| boat | 0.73 | 0.43 |
| fire-hydrant | 0.74 | 0.35 |
| airplane | 0.81 | 0.19 |
| cow | 0.73 | 0.24 |
| backpack | 0.39 | 0.3 |
| cup | 0.46 | 0.23 |
| surfboard | 0.66 | 0.21 |
| tie | 0.26 | 0.23 |
| sports-ball | 0.47 | 0.12 |
| kite | 0.57 | 0.19 |

Table 6: Values for $\tau$ used for the gist criterion throughout the paper (test set).

| Task | Hard Positives | Hard Negatives |
|---|---|---|
| car | 0.47 | 0.5 |
| bowl | 0.65 | 0.57 |
| boat | 0.72 | 0.42 |
| fire-hydrant | 0.76 | 0.35 |
| airplane | 0.81 | 0.19 |
| cow | 0.71 | 0.23 |
| backpack | 0.39 | 0.31 |
| cup | 0.46 | 0.23 |
| surfboard | 0.68 | 0.21 |
| tie | 0.25 | 0.23 |
| sports-ball | 0.44 | 0.12 |
| kite | 0.58 | 0.19 |

Table 7: Values for $\tau$ used for the gist criterion throughout the paper (validation set).

| Method | Hyperparameter Name | Values |
|---|---|---|
| GDRO | $K$ | [5, 30, 60, 240] |
| IRM | $\lambda$ | [0.1, 1, 3, 10] |
| CVaR | $p$ | [0.05, 0.1, 0.15] |
| Focal | $\gamma$ | [0.2, 0.5, 0.7, 1] |
| Label Reweighting | $\alpha$ | [0.2, 0.5, 0.7, 1] |
| Label Undersampling | $\alpha$ | [0.2, 0.5, 0.7, 1] |
| Environment Reweighting | $\alpha$ | [0.5, 1, 1.5, 2] |
| Environment Undersampling | $\alpha$ | [0.2, 0.5, 0.7, 1] |

Table 8: Hyperparameter lists for each method (ERM has no method-specific hyperparameters).

## C  Experimental Details

### C.1  Licences

The Resnet-50 is provided under an Apache 2.0 licence.

### C.2  General Hyperparameters

We train (where not otherwise indicated) using SGD with learning rate of 1e-4, momentum of 0.9, L2 weight regularization of 1e-4, and batch size 32 (the largest we could fit on the available GPUs). For all methods, we use early stopping on the validation loss with a patience of 3 epochs. All images were resized to $321 \times 321$.

**Training Parameters for ERM in Sec. 3.**  For the experiments which refer to all 171 tasks (rather than just the 12 we focus on in NOOCH , we use an Adam optimizer with 1e-3 learning rate and batch size 16. All other training parameters are the same as noted elsewhere.

**Sentence Embeddings.** To computer sentence embeddings, we use `stsb-distilroberta-base-v2` from the `sentence_transformers` package (https://www.sbert.net/).

### C.3  Compute

We used GPUs on internal clusters, utilizing Nvidia Titan XP, T4, and P100 GPUs. The total number of jobs run to produce our main results, i.e. Table **??** and all associated figures, is

$$\text{num\_tasks} \times \text{num\_seeds} \times \text{num\_methods} \times \text{num\_hyperparameters\_per\_method} \quad (1)$$
$$= 12 \times 3 \times 5 \times 3.2 = 576 \quad (2)$$

where we calculate num_hyperparameters_per_method from Table 8. The job runtime lengths varied widely due to the use of early stopping.

### C.4  Method-Specific Hyperparameters

Each of the methods we run other than ERM has a hyperparameter which trades off better overall performance and OOC performance. For each method, we swept over 4 values of these parameters (3 for CVaR), running 3 seeds at each value. The values of these hyperparameters are shown in Table 8. They were chosen through some manual experimentation to get a sense of the useful ranges for each.

### C.5  Classification Error Results for Experimental Section

In Sec. 6, we show results using AUC, NLL and ECE metrics - here we show analogous plots for classification error.

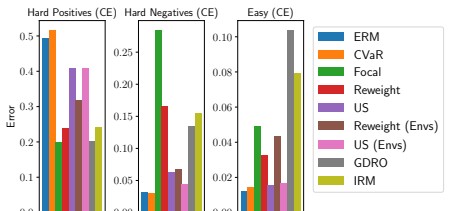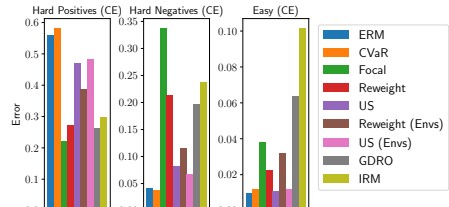

Figure 14: Classification error achieved on hard positive, hard negative, and easy examples.

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

Table 9: AUC on hard test examples for all 12 NOOCH stress tests, after hyperparameter selection. Bold numbers have overlapping standard deviations. L: NOOCH-CE. R: NOOCH-Gist.

## C.6 Detailed Results

For each metric of AUC, NLL, Error, and ECE, each of the 12 tasks in NOOCH , and both NOOCH-CE and NOOCH-Gist, we show results below with standard deviation across 3 seeds. There are the same results as Figures 4, 14, 5, are 6, but shown in more detail.

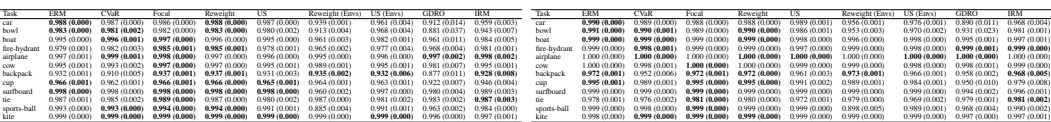

Table 10: AUC on easy test examples for all 12 NOOCH stress tests, after hyperparameter selection. Bold numbers have overlapping standard deviations. L: NOOCH-CE. R: NOOCH-Gist.

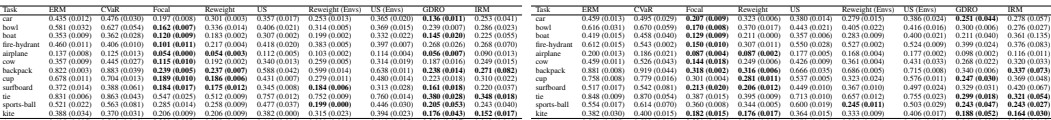

Table 11: Classification error on hard positive test examples for all 12 NOOCH stress tests, after hyperparameter selection. Bold numbers have overlapping standard deviations. L: NOOCH-CE. R: NOOCH-Gist.

Table 12: Classification error on hard negative test examples for all 12 NOOCH stress tests, after hyperparameter selection. Bold numbers have overlapping standard deviations. L: NOOCH-CE. R: NOOCH-Gist.

Table 13: Classification error on easy test examples for all 12 NOOCH stress tests, after hyperparameter selection. Bold numbers have overlapping standard deviations. L: NOOCH-CE. R: NOOCH-Gist.

Table 14: NLL on hard positive test examples for all 12 NOOCH stress tests, after hyperparameter selection. Bold numbers have overlapping standard deviations. L: NOOCH-CE. R: NOOCH-Gist.

Table 15: NLL on hard negative test examples for all 12 NOOCH stress tests, after hyperparameter selection. Bold numbers have overlapping standard deviations. L: NOOCH-CE. R: NOOCH-Gist.

Table 16: NLL on easy test examples for all 12 NOOCH stress tests, after hyperparameter selection. Bold numbers have overlapping standard deviations. L: NOOCH-CE. R: NOOCH-Gist.

Table 17: ECE on hard test examples for all 12 NOOCH stress tests, after hyperparameter selection. Bold numbers have overlapping standard deviations. L: NOOCH-CE. R: NOOCH-Gist.

**L: NOOCH-CE**

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

Table 18: ECE on easy test examples for all 12 NOOCH stress tests, after hyperparameter selection. Bold numbers have overlapping standard deviations. L: NOOCH-CE. R: NOOCH-Gist.

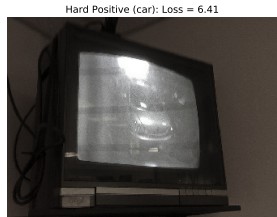 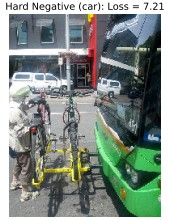 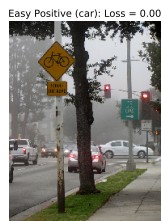

Figure 15: Examples from the `car` task.

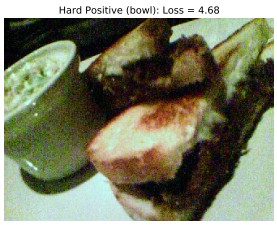 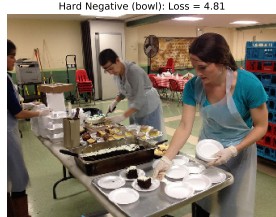 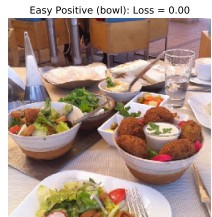

Figure 16: Examples from the `bowl` task.

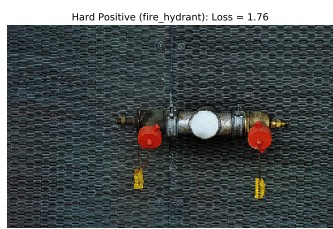 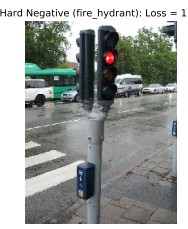 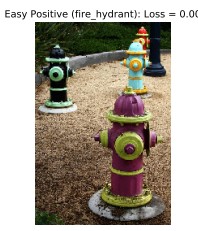

Figure 17: Examples from the `fire_hydrant` task.

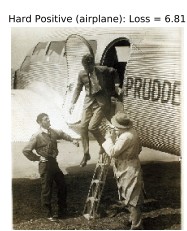 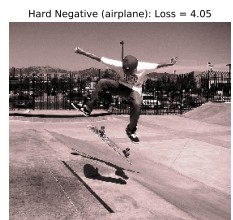 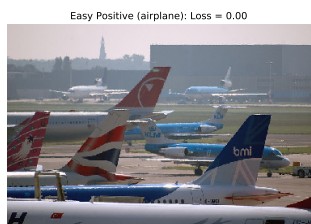

Figure 18: Examples from the `airplane` task.

# D  Qualitative Analysis

## D.1  Examples

Here we show examples from NOOCH-CE: from left to right we show a hard positive, hard negative, and easy positive for each of the 12 tasks. We show the loss for a sample ERM model in the title of the caption. We choose these examples by finding the highest (lowest) loss ERM examples for each hard (easy) category — additionally, for ease of viewing, for hard positives, we choose examples for which the target object's area is larger than average for positive examples of that class. We note that label noise can be a problem in COCO, as in any dataset, and finding the highest loss examples can occasionally surface a mislabelled datapoint (see Fig. 23, the Hard Negative in the `tie` class, where a tie is present but is not labelled). Along with subjectivity in labels, mislabelling is an occasional hazard of any dataset requiring mass labelling — fortunately, we did not find too many occurrences of mislabelled points when examining hard positives and negatives in NOOCH.

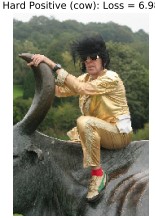
Hard Positive (cow): Loss = 6.98

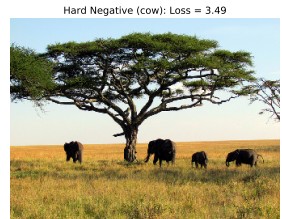
Hard Negative (cow): Loss = 3.49

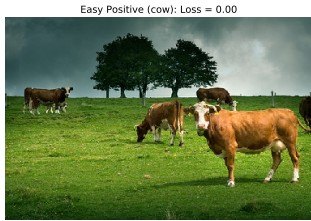
Easy Positive (cow): Loss = 0.00

Figure 19: Examples from the `cow` task.

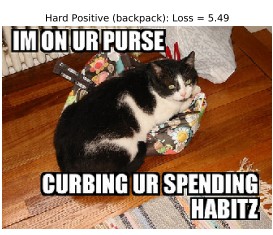
Hard Positive (backpack): Loss = 5.49

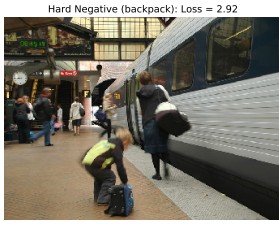
Hard Negative (backpack): Loss = 2.92

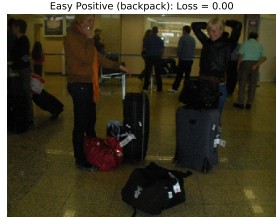
Easy Positive (backpack): Loss = 0.00

Figure 20: Examples from the `backpack` task.

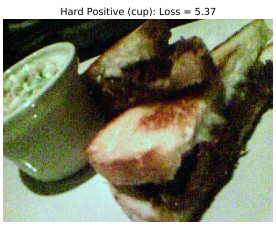
Hard Positive (cup): Loss = 5.37

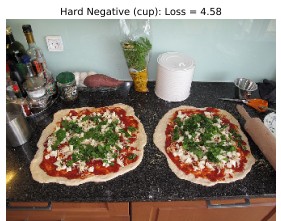
Hard Negative (cup): Loss = 4.58

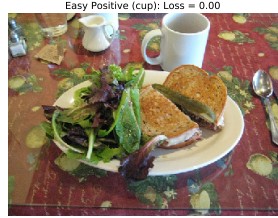
Easy Positive (cup): Loss = 0.00

Figure 21: Examples from the `cup` task.

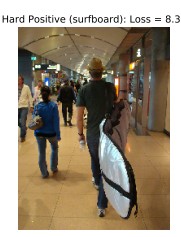
Hard Positive (surfboard): Loss = 8.32

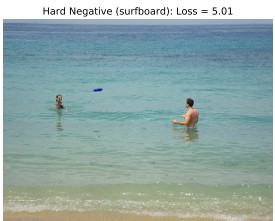
Hard Negative (surfboard): Loss = 5.01

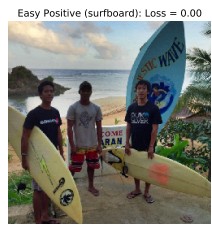
Easy Positive (surfboard): Loss = 0.00

Figure 22: Examples from the `surfboard` task.

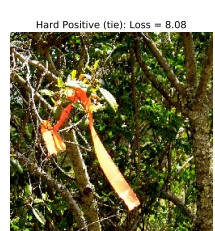
Hard Positive (tie): Loss = 8.08

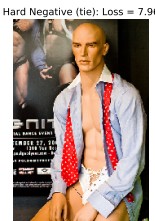
Hard Negative (tie): Loss = 7.96

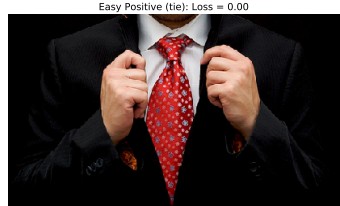
Easy Positive (tie): Loss = 0.00

Figure 23: Examples from the `tie` task.

## D.2    Analyzing GDRO's Errors

In Fig. 7, we show the images with the largest gaps in NLL on the `car` task among two groups: hard positives and hard negatives (in NOOCH-CE), where GDRO is better than ERM. Here, in Fig. 26 we look at the images where GDRO most underperforms ERM. These help us consider some pitfalls

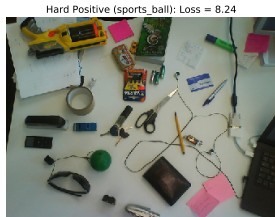 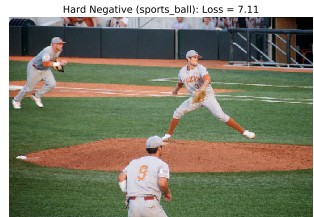 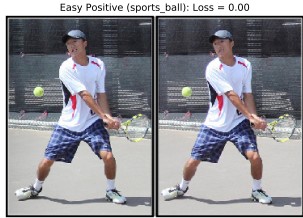

Figure 24: Examples from the `sports_ball` task.

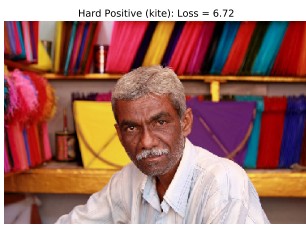 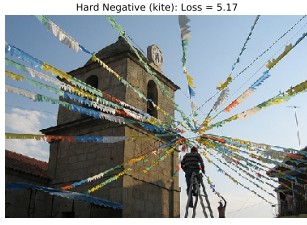 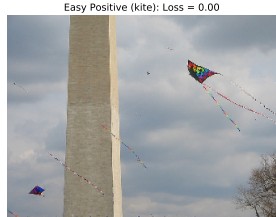

Figure 25: Examples from the `kite` task.

of the GDRO objective, which incentivizes performance on two small subgroups: "cars without roads" and "roads without cars". GDRO's worst hard positive is reminiscent of a "road without a car": the background (labelled "pavement") looks road-like but the focal object is not a prototypical car (the car is in the top left, background). We hypothesize that GDRO's incentive to perform well on "roads without cars" pushed its prediction negative, whereas ERM was able to take advantage of context and correctly output a positive. GDRO's worst hard negative tells an opposite story: the snowy background looks *less* road-like, but the stop sign and black objects in the snow suggest a car may be present. We hypothesize that GDRO's incentive to perform well on "cars without roads", may have pushed its prediction wrongly *positive* in this instance.

Qualitatively, it appears that GDRO uses context in potentially more unintuitive ways than ERM does. While we hope that robust methods will be more interpretable since they may ignore irrelevant signals in the training data [22, 43, 51], these two goals may sometimes be at odds. Indeed, we observe that robust objectives may *unnecessarily* encourage predictions which diverge from the context.

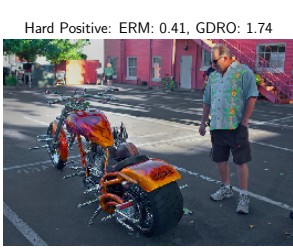

Figure 26: Test set examples with largest difference in NLL between ERM and GDRO methods, where ERM is better, among (L) hard positives and (R) hard negatives from the NOOCH-CE `car` category. Titles show NLL for each method on that image.