# OpenReview forum: "Identifying and Benchmarking Natural Out-of-Context Prediction Problems"
_NeurIPS.cc/2021/Conference — NeurIPS 2021 Poster_

### Official Review · Reviewer_h5rh · 2021-07-15

**Rating:** 4
**Confidence:** 4

**Summary:**

This work presents two novel approaches automatically identify out-of-context instances for image classification tasks, based either on co-occurences with other object classes or the overall gist of a scene. The authors show that these two criteria are useful at automatically detecting challenging OOC instances and evaluate the generalization ability of a number of robust learning methods on "hard" and "easy" splits of the dataset.

**Ethical Concerns:**

I do not believe there are ethical concerns with this work.

**Limitations And Societal Impact:**

The authors have effective communicated the limitations of their work, however they have not discussed potential societal impacts.


**Main Review:**

Evaluating and improving model performance on out-of-context settings is an important task machine learning research, and this work shows a feasible way to automatically detect naturally occurring OOC examples in the MS-COCO dataset based on information available from MS-COCO, COCO-Stuff, and a (pre-trained) SBERT model. The originality of this work lies in the manner the sources of external information are leverage to automatically define a standard context for and object class. Based on this established standard context hard-negatives and hard-positives could be clearly defined based on their relationships with the standard context.

The central components to the proposed approaches are presented clearly in this work, however the takeaways from this research are less clear. The results indicate the researchers have successfully identified harder and easier examples from the datasets of interests, but do not contrast these results with other methods of determining hard-negative/hard-positives. The proposed approaches also seem highly specialized to the COCO dataset and it's unclear how these approaches could be effectively applied to other vision datasets, this also impacts the significance of the findings.  There are several tests of how various robust learning approaches fare on the proposed benchmarks, which has interesting results for calibration and generalization performance, however a comparison of the observed benchmark with other OOC benchmark test would further demonstrate the significance of the proposed automatic detection of OOC instances.

--- clarity in presentation of the graphs could be improved by ensuring that paired graphs share the same y-axis. ---


**Time Spent Reviewing:**

6

---

> ### Author Response · Authors · 2021-08-06
> **Response: a comparison to another OOC benchmark, significance, and annotations**
>
> Thank you for the constructive feedback - we agree that we can do more to make the takeaways of our research clearer; we will try to do so in this response as well as in the next version of the paper.
>
> ### A New Experiment Comparing NOOCH to another OOC benchmark
>
> Firstly, we agree with you that a comparison to another OOC benchmark is an important piece of empirical work currently missing from the paper, which would help establish motivation. To this end, we have performed such a comparison, running all of the same methods on the Waterbirds dataset, which contains a semi-synthetic shift in image background specified by environment variables. This experiment is discussed in depth in the Overall Response, but we will give a summary here. The results on the Waterbirds dataset are shown below, with similar results on worst-environment error rate.
>
> AUC on Hard Examples
>
> * ERM: 0.691 (0.048)
> * GDRO: 0.929 (0.005)
> * IRM: 0.788 (0.015)
> * CVaR: 0.703 (0.016)
> * Focal: 0.506 (0.080)
>
> In short, we find that running on Waterbirds, a semi-synthetic dataset whose shift is specified exactly by environment variables, produces a more favourable evaluation of environment-based methods (in particular, GDRO) than running on NOOCH, a non-synthetic dataset where the environment variables are less cleanly related to the shifts of interest. This suggests that synthetic datasets may overrate the practical performance of environment-based methods, and motivates the creation of non-synthetic datasets such as NOOCH to evaluate similar phenomena. We found this experiment very enlightening and plan to include it in the next version of the paper.
>
> ### Significance
>
> We are a little unclear on what you mean by “other methods of determining hard-negative/hard-positives”. The method that first comes to mind is something like Imagenet-A, where we would choose the examples which have highest loss across some ensemble of models. While we agree that this would indeed produce a set of examples with very high loss (likely higher loss than those in NOOCH), we think that NOOCH still provides value due to the way it explores various important failure modes of ML models (e.g. spurious correlations). Importantly, NOOCH focuses on the OOC problem whereas the images in Imagenet-A may be hard for any number of reasons; we think that our work provides some valuable interpretable understanding about possible mechanisms of model failure. We discuss this further in the Overall Response. We agree that this aspect of our motivation is under-addressed in the paper currently and will add more discussion in the next version.
>
> ### Annotations
>
> Finally, we address the question of how “specialized” our results are to COCO/COCO-Stuff. In the Overall Response, we discuss our usage of annotations in depth. We note that the full dataset does not require annotations for the type of method we propose to be successful, rather just some large-enough evaluation set. To this end, many datasets could be adapted to suit our proposed methods without too high of a cost. Further, we hope that our ideas presented here will inspire many future methods along these lines for practitioners working with their own applications and their own forms of available auxiliary data, who wish to systematize their robustness evaluations. Finally, we note that even if the specific types of annotations available in this dataset are not currently available in most others, the proposed benchmark is still useful for evaluating proposed robustness/domain generalization methods.
>
> Thank you for the data visualization feedback, we will adjust accordingly. We will also hope to address questions about societal impacts more fully, for instance the tradeoffs between performance on small subgroups and issues of privacy/surveillance. Let us know if you had other questions in mind surrounding societal impacts.

---

> > ### Comment · Reviewer_h5rh · 2021-08-31
> > **Response to rebuttal**
> >
> > Thank you for your clarifications in response to this review as well as the others in this paper, especially including the additional dataset for evaluation.
> >
> > Overall, the discussion on this work has improved my understanding of the proposed approaches and how they may incorporate with existing literature and out-of-context problems. My understanding of NOOCH has shifted from understanding this paper as an automatic technique that could be applied to a number of vision datasets, to something closer to the development of new dataset to evaluate robustness and a prescriptive description of the dataset design which could be used in similar settings. While this may not be fully automatic, that isn't the main goal or contribute and doesn't detract from the value of this work or how it may be built upon.
> >
> > I plan to adjust my scoring based on this.

---

> > > ### Author Response · Authors · 2021-09-09
> > > **Thank you**
> > >
> > > Thanks for your response! We're glad the rebuttal was helpful for improving your understanding of the paper, we appreciate the feedback and plan to incorporate it heavily in future versions of the paper.

---

### Official Review · Reviewer_25HS · 2021-07-16

**Rating:** 5
**Confidence:** 3

**Summary:**

The paper suggests that it is possible to extract examples from existing computer vision datasets that represent “out-of-context” (OOC) examples, i.e. examples that differ in contextual information from that in most of the images (per class), either through the absence of commonly co-occuring objects (CE criterion) or unusual scene-based deviation (gist criterion).

In MS-COCO, using instance-labels and segmentation-mask-areas, images are identified that are likely to exhibit potential object co-occurence based confusion (if a pair of objects co-occur at a high rate, and the co-occuring object tends to occupy a larger spatial area). If the caption-embeddings for an image differ beyond a threshold from the average embedding for the class of interest, then the image is said to differ in a “gist shift” sense. Hard positives are such context-shifted examples, and hard negatives are examples where the common context appears but not the object of interest. 12 classes are identified that seem to suffer the most based on differences in NLL with trained binary classifiers.

An empirical comparison is performed across existing OOD generalization methods for binary classification tasks on the identified subsets, showing mixed results across all subsets and evaluation metrics.

**Limitations And Societal Impact:**

Limitations and societal impact have been adequately discussed.

**Main Review:**

Since the paper’s primary contribution is to introduce a new OOD generalization dataset in the presence of many existing ones (some of which are identified by the paper), it is necessary to provide a compelling motivation. The paper states that the proposed sets/tasks are “a closer analogue to the OOC problems that arise in the standard model development cycle” and therefore “might more scalably provide insights into the challenges faced by current approaches”.

 - This motivation needs clearer unpacking, because it is not clear to me that a set of binary classification tasks extracted from a much larger multiway classification dataset with heuristics such as using “an object’s size within an image as a proxy for extractibility“ is any more a closer analogue to problems that “naturally arise” than alternatives such as splitting full-scale datasets by environments, as done for WILDS [1], or curating a collection of examples that trained classifiers fail to perform well on, as done for Imagenet in Imagenet-A [2].

 - The method for creating stress-tests presented in the paper would not be applicable generally for any computer vision dataset in its “model development cycle”, since one would require costly per-object annotations (segmentation masks) and image captions, which don’t exist for most computer vision datasets currently.

 - The binary classification splits might be a way to whittle down to the 12 chosen classes, but typically people perform multi-way classification with MS-COCO. Perhaps keeping the challenge-task and evaluations multi-way might be more aligned with the paper’s stated goal of developing a “naturally-arising” OOC problem.

Minor:

L201: “compliment” —> “complement”

L278” “Fig. 11” —> “Fig. 8”

L327: “what we” —> “that we”

On the whole, I think the paper could be improved with stronger motivations for the proposed test sets and tasks, and demonstrations of wider applicability of the method used to develop such OOC stress-tests when it is not always within budget to acquire annotations as detailed as in MS-COCO.

[1] WILDS: A Benchmark of in-the-Wild Distribution Shifts, 2020

[2] Natural adversarial examples, 2019

--------------
I have read the rebuttal, and as of now stand by my rating.


**Time Spent Reviewing:**

4

---

> ### Author Response · Authors · 2021-08-06
> **Response: on related work, annotations, and binary vs. multi-label classification**
>
> We thank the reviewer for their constructive feedback - we agree that the points raised are important ones and we hope to clarify some important strands in our response.
>
> ### Related Work
>
> In the Overall Response, we discuss in depth the relationship between NOOCH and the related work you bring up - we will quickly summarize here. First, we run a new experiment where we compare the performance of all methods on Waterbirds, a semi-synthetic OOC-type dataset whose (synthetic) shift is specified precisely by environment variables. We find that environment-based methods tend to perform vastly better than ERM on the synthetic shift in Waterbirds, whereas they perform only marginally better than ERM on NOOCH. This adds empirical motivation to our work - we find that synthetic shifts may be vulnerable to relatively over-estimating the performance of environment-based methods, demonstrating the need for a non-synthetic, curated dataset such as NOOCH. This experiment will be included in the next version of the paper.
>
> We also discuss our motivation conceptually in the Overall Response. At a high level, we hope to clarify the motivation behind our work as follows: NOOCH focuses on specifying and exploring interpretable classes of model failure at the example level. This contrasts with something like WILDS, which does an excellent job of providing difficult, real-life distribution shifts but does not necessarily provide any insight as to the types of failures which occur. Similarly, while Imagenet-A finds examples with high loss, it is hard to draw any systematic conclusions from a model’s poor performance on that dataset. We agree that this point about interpretable mechanisms of model failure is an important one for NOOCH’s motivation, and agree that it should be drawn out more in the paper - this will be improved for the next version.
>
> ### Annotations
>
> We also discuss the question about annotations in the overall response: one key point is that the entire training set does not need annotations, only an evaluation set. This cost may not be too high, particularly in comparison to the cost of other methods of checking model performance out-of-context, which may be human-driven and somewhat ad-hoc.
>
> ### Binary vs. multi-label classification
>
> We agree that the binary classification task that we propose is not the one originally intended in the COCO challenge. We chose this task in order to stay in line with the recent domain generalization literature, which tends to focus on binary classification (e.g. [1, 2]). We note that the standard COCO tasks are closer to multi-label classification rather than multi-way, since there may be multiple objects of interest in each image. To this end, we perform the following comparison. We ran ERM models on the multi-label classification task - classifying the presence or absence of all 171 COCO-Stuff categories at once. We find that the AUC achieved on hard NOOCH examples by ERM in the multi-label setting is comparable to that in the binary setting - the multilabel classifier is slightly worse on all tasks and criteria, by amounts varying from 2e-3 to 0.05. We would expect ERM to be a little worse in the multilabel setting since it is trying to optimize for many tasks at once; the question of how to best use extra labels as auxiliary information is indeed a fascinating direction for future work. It would be nice to further compare GDRO/IRM to ERM on the multi-label task; however, it is not clear how to define the environments appropriately when there are a large number of tasks of interest, so this also seems like an interesting direction for future work. Nonetheless, we hope that this experiment shows that we do not lose too much in this benchmark by moving from a multilabel to a binary setting.
>
> Thank you for the typo corrections, we will address these.
>
> [1] Sagawa et al. Distributionally Robust Neural Networks for Group Shifts: On the Importance of Regularization for Worst-Case Generalization.
>
> [2] Arjovsky et al. Invariant Risk Minimization.

---

> > ### Comment · Reviewer_25HS · 2021-08-22
> > **Thanks for the response**
> >
> > Responses about Related Work:
> >
> > I’m in agreement that synthetic-shift-datasets aren’t the best way to go. But my comments weren’t about such benchmarks, they were about more “practical” benchmarks where it also appears that environment-based invariance methods struggle. Regarding interpretability, this is quite entangled with my comment about annotations; at the moment the proposed benchmarks only enable very specific interpretability and any attempt to extract similar insights in different domains/datasets would be bottlenecked by the cost of seeking annotations, which need to be informed of the nature of spuriousness we expect in the problem.
> >
> > Responses about Annotations:
> >
> > I’m not sure what’s being considered as “human-driven and ad-hoc” when referring to “other methods of checking model performance”, but I’d argue that the principle behind NOOCH, if applied to anything beyond the setting explored in the paper, also potentially lends itself to such description, since we need to know what sort of spuriousness we expect and try to find splits of data accordingly after coming up with ways to solicit annotations reflecting such expectations.
> >
> > Response about binary/multi-label:
> >
> > I’m not sure why there is any need to “stay in line with the recent domain generalization literature, which tends to focus on binary classification”. The methods themselves rarely require a binary classification setting (the two cited papers [1,2] definitely do not), and while such binary classification toy-datasets have often been used for experiments in such papers, that should not be of concern to a submission whose stated goal is to develop a benchmark reflecting “naturally-arising” problems.

---

> > > ### Author Response · Authors · 2021-08-23
> > > **Thanks for your response!**
> > >
> > > Thank you for the response - hopefully we can address some of your followup questions here.
> > >
> > > **Related work**: We note that the benchmarks you reference, WILDS and Imagenet-A, both require costly “annotations” to assemble. WILDS presupposes the existence of relevant annotations for creating environments: for example, timestamps, locations, information about the race/gender of an input’s topic. The creation of Imagenet-A requires a different type of “annotations” - the outputs of an ensemble of ResNets. Neither of these are guaranteed to exist in a dataset - for instance, nothing of that type is present in MS-COCO. Timestamps may in fact be impossible to collect after training time, and training an ensemble may be computationally resource-intensive. We hope to demonstrate with our paper that this type of process (followed implicitly by WILDS/Imagenet-A) can be abstracted; the existence of some auxiliary annotations (timestamps, ensemble outputs, captions, segmentations, something else) can allow us to reason about OOC-ness in a potentially rich way.
> > >
> > > **Binary/Multi-label**: We agree that, despite the ML research field’s focus on classification, there are a number of more relevant tasks to industrial practice, as broadly ranging as ranking, structured prediction, or uncertainty-aware classification. One thing that may help here is drawing a distinction between the OOC problem (identifying cows when they are not on a grassy field) and the task (binary classification of cows). Our paper is concerned with identifying OOC problems, which may cause issues across a range of tasks, and we present binary classification tasks which may help probe these. If you think this is a helpful distinction, we are happy to specify the scope more carefully in our paper in such a way.
> > >
> > > We believe that the OOC problems which “naturally arise” in a binary classification setting will also be problematic in a more structured/complex task. Therefore, we consider the OOC problems identified to be “naturally-occurring”, even if there are more complex tasks which could be of interest as well. As a proof of concept, we show experimentally in our prior response that the OOC problems identified in the binary classification setting also transfer to the multilabel setting.
> > >
> > > **Annotations/Ad-Hoc-ness**: We agree that it would be great to have a method which could surface any type of issue that a model experiences with OOC examples automatically. We consider that to be a natural goal of this line of research, and a worthy direction for future work. Such a method would be like a program which automatically “debugs” an ML model; we consider our work to be a useful first step in that direction, by 1) formulating the problem and 2) demonstrating how one can move towards that goal by generating “test cases” using auxiliary data which might be present. We think this is a strength of our "know what you're looking for" approach - analogously to writing test cases, we are testing for plausible flaws which might be present and can draw systematic conclusions from their outputs, which may be able to direct improvements to the modelling process.
> > >
> > > By “ad-hoc” we mean the following: suppose one is worried about the vulnerability of a trained model to spurious correlation. One can manually sort through examples to see if examples where the correlation is present are predicted incorrectly - for instance, possibly using the heuristic of looking at the highest loss examples. It is our impression that at the moment, there is no abstracted, scalable version of this process.

---

### Official Review · Reviewer_LYwb · 2021-07-16

**Rating:** 8
**Confidence:** 4

**Summary:**

The authors propose an approach to identify "naturally" occurring out of context prediction problems within an existing dataset, focusing specifically on COCO. Using their found images, they report robustness results from a range of models and propose a future challenge task.

**Limitations And Societal Impact:**

One potential societal impact to consider: OOC images might be more likely to inherently represent minority populations or viewpoints. Are there potential consequences from proposing a method to actively surface these?

**Main Review:**

Originality: to my knowledge, this is the first generalizable approach to automatically finding existing OOC images from datasets. Existing work seems well-cited.

Quality: the submission seems technically sound, and the claims seem supported by the author's evaluations. I think the proposed approach seems interesting and insightful, and appreciate the depth of the discussion the authors provided about its nuances and limitations. I would have liked to see the authors discuss in more detail the practical matter of creating challenge benchmarks from these images that they discover. Is there a standardized way that results should be reported?

Clarity: this submission is well-written and organized. The motivation is clearly discussed.

Significance: This work seems of strong importance to the neurips community. That said, one potential area for improvement would be for the authors to better establish the significance of this work. Why is it so important to identify natural OOC examples versus synthetic ones? Can we better improve models by having real examples like this? If so, how?

**Time Spent Reviewing:**

2 hours

---

> ### Author Response · Authors · 2021-08-06
> **Response: on synthetic examples and results reporting**
>
> Thank you for the feedback! We’re pleased that you found our work to be interesting and insightful, and agree that the problem is an important one.
>
> ### Synthetic Examples
>
> We address your question re: synthetic data in the overall response, where we compare the results of running the same methods on Waterbirds (a dataset with a synthetic shift described cleanly by environment variables) to the results on NOOCH. We find that on this semi-synthetic data, the environment-based methods vastly outperform the others, in contrast to NOOCH, where the environment-based methods only slightly outperform. This suggests that when using synthetic data, methods which are given access to extra information about the synthetic shift may receive a more favourable evaluation; in contrast to a less-synthetic dataset like NOOCH, where the shift is less well-specified, and the extra environment information is only marginally helpful. This gets at a larger question: how can we systematize OOC evaluation without abstracting away the real-life messiness which makes OOC prediction so hard? To this end, the goal of our paper is to move towards a situation where we can formally specify certain notions of OOC-ness, while maintaining a realistic picture of what it means to be OOC.
>
> ### Results Reporting
>
> We agree that results reporting is extremely important. However, we are not sure there is a one-size-fits-all solution for every dataset, given the vast difference in potential datasets. For instance, in the case of NOOCH, we found that AUC was a useful metric due to the label imbalance in COCO-Stuff. However, in more balanced datasets it may make more sense to look at error rate, or potentially a weighted error rate depending if false positives/negatives are more costly. In some cases, even the ideas of positives/negatives may not be as clear (for instance, regression tasks). We would hope that in whatever the application of interest is, results are reported across a number of different metrics of relevance since this gives a clearer picture of model performance. Additionally, it is not necessary that others create benchmarks out of their datasets of interest; rather our main hope is that these ideas can help systematize evaluation of robustness for their target applications (although of course sharing the data/results is always ideal).
>
>
> We appreciate the thoughtful questions and hope to address them more fully in the next version of the paper. We will also hope to address some of your questions re: societal impact - as with any question around including outliers or minority groups in data, there comes a tradeoff between representation and surveillance/"machine readability". It certainly seems that for some sensitive applications we may prefer to use smaller, less-representative datasets (or limit the application of ML at all), but for some, under-represented populations prefer to receive high-quality performance from the model.

---

### Official Review · Reviewer_wXHX · 2021-07-16

**Rating:** 8
**Confidence:** 3

**Summary:**

This paper presents a method to find hard, out-of-context examples within a training set by using existing bounding box annotations.

**Limitations And Societal Impact:**

Limitations and social impact are adequately discussed.

**Main Review:**

This paper tackles a very important question of understanding model failures, particularly on hard, out of distribution, or atypical examples.

The authors propose a novel definition for identifying out of context examples by using existing bounding box annotations for an image, and categorizing the relationships between various objects across the dataset. The method is unique, and qualitative results are interesting.

The authors also perform a thorough benchmarking analysis of the performance of ERM as well as other robust-learning alternatives, and analyze the results using their out of context prediction definition.

Writing exposition is very clear.

Comments:
- Can this method only be used if a dataset has complete bounding box annotations? That seems very costly for most use cases.
- What is the intuition behind using -CE or -Gist? A greater discussion of the practical tradeoffs & settings where each is preferrable would be appreciated.

The method is very clear, and perhaps may provide inspiration for designing out-of-context definitions in other settings.

**Time Spent Reviewing:**

2

---

> ### Author Response · Authors · 2021-08-06
> **Response: On Annotations and Intuition**
>
> Thank you for the positive response! We are glad that you found our work insightful and clear.
>
> ### Annotations
>
> We discuss your question re: annotations in the overall response at length. Here, we note that only an evaluation set requires annotations, rather than the full training set, which may not be too expensive. We also note that in general, verifying the robustness of a model to OOC/OOD data (to deployment standards) may be an expensive task! The methods proposed in this paper may require collecting annotations, but are at least a step in a more systematic direction.
>
> ### Intuition
>
> We note that we do discuss the intuitions behind CE and Gist in the paper (see Line 66-72 and Line 91-98 respectively). Additionally, we discuss some examples to contrast the two criteria in Fig 2 and lines 133-140, as well as Fig 9 and lines 284-301. We are happy to add some further discussion in the paper - if you have any specific questions you’d like us to answer please let us know! We’ll also quickly re-iterate some intuition from the paper here.
>
> For CE: as noted in a number of works (a good discussion in Lovering et al [34]), two important factors for how likely a model is to pick up on a signal in the dataset is 1) how much it co-occurs with the label, and 2) how “extractible” it is (how easily learned). We can calculate the co-occurrence from the data, and take an object’s area to be a proxy for extractibility. For Gist: we make the assumption that a caption is a good proxy for the gist of the scene. A number of works have shown that distances in sentence embedding space are useful for determining similarity - we make the assumption that if images have captions which are far away (in embedding space) from average captions for this object, this images gist is also unusual. One way to think about the tradeoff between these two is that CE is more targeted - it focuses on particular co-occurring objects which may not be present (for instance, the canonical “cow on the beach” example - if a model learns to recognize a cow on grass, it will have difficult if it is on sand). Gist, on the other hand, focuses on unusual scene configurations rather than the presence or absence of one object in particular.

---

> > ### Comment · Reviewer_wXHX · 2021-08-06
> > **Thank you for the clarifications.**
> >
> > Thank you for the clarification regarding CE & Gist.
> >
> > For the bounding box annotations - the fact that we only need them during evaluation is noted. I agree that verifying robustness may be an expensive task. However, in this case, the bounding box annotations are used to generate a relatively coarse signal - co-occurence between objects and relative sizes of objects. It is unclear if detail & precision at the level of bounding box annotations are necessary to measure these attributes. I agree that if they exist, bbox annotations are indeed a useful signal. However, if we were to evaluate a model on a new data set or data stream, I am still not convinced that bbox annotations are the most cost effective method here. Nonetheless, this paper provides a great proof-of-concept at the very least of how such information could be used, and it is perhaps not the authors' job to find the most cost effective method (although a discussion of these drawbacks would certainly help).

---

> > > ### Author Response · Authors · 2021-08-09
> > > **Bounding Boxes**
> > >
> > > Yes definitely! There definitely could be a less work-intensive way of formulating this signal - since bounding boxes/segmentations are more commonly found than other potential proxies for extractability, we went with those. We agree this functions as either a proof of concept or an effective method for using "found" information, and we're happy to discuss some of these tradeoffs in the next version.

---

### Author Response · Authors · 2021-08-06
**Overall Response: on 1) the significance of our contribution and 2) our use of annotations**

We thank all the reviewers for their thoughtful commentary and constructive feedback. In this overall response, we address two common concerns: on the significance of our contribution relative to other OOC-type benchmarks, and on our usage of annotations.

## Significance of Our Contribution

We appreciate the reviewers’ questions about the significance of our contribution relative to existing OOC-type datasets - we hope to make this clearer both with 1) added experiments and by 2) drawing additional connections between various methods of finding OOC examples in the literature.

### 1. Experimental Comparison

First, we take Reviewer h5rh’s suggestion of comparing NOOCH with an existing OOC benchmark, in order to highlight the novel insights produced by NOOCH. We compare with the Waterbirds benchmark, recently proposed by Sagawa et al [1] for testing models’ robustness to spurious correlations. It is a good example of a semi-synthetic OOC benchmark as referred to in the paper: each example image contains a bird from CUB pasted onto a background from the Places dataset. The classification task is to identify if the bird is a waterbird or a landbird. At training time, waterbirds are mostly shown on water backgrounds, and landbirds are mostly shown on land backgrounds; at test time, this is reversed (e.g. waterbirds are mostly shown on land backgrounds). This defines the 4 environments: landbirds on land, waterbirds on land, etc.

We show below the results of running each method on Waterbirds (with standard deviations over 3 seeds shown, and a similar hyperparameter selection process to the one used in our paper). We show results for two metrics: worst-environment error (as in [1]) and AUC on hard examples (defined as the two environments where the bird and background do not match, e.g. landbird on water).  We find that the environment-based methods (GDRO/IRM) perform vastly better on Waterbirds than the other methods. In contrast, on NOOCH, these methods offer only marginal improvement. GDRO improves over ERM by 0.2 AUC on Waterbirds, and by only 0.02-0.04 AUC on NOOCH - there is an order of magnitude difference between the performance gaps on these two datasets.

Worst-Environment Error (as in [1])

* ERM: 0.400 (0.050)
* GDRO: 0.108 (0.004)
* IRM: 0.127 (0.015)
* CVaR: 0.354 (0.002)
* Focal: 0.402 (0.012)

AUC on Hard Examples

* ERM: 0.691 (0.048)
* GDRO: 0.929 (0.005)
* IRM: 0.788 (0.015)
* CVaR: 0.703 (0.016)
* Focal: 0.506 (0.080)

This result has an intuitive interpretation: Waterbirds, which contains a synthetic shift defined precisely by the environment variables, provides a more favourable evaluation of methods that utilize the information from these environment variables. On the other hand, the environment-based methods yield less of an improvement on the hard examples identified by NOOCH, whose shift is less well-specified. This suggests that by using real examples found in an existing dataset rather than creating a synthetic shift, NOOCH may provide a more realistic picture of current environment-based methods’ performance in practical scenarios, where environments may not describe the relevant context shift so cleanly.

We will include this experiment and analysis in the next version of the paper.

### 2. Conceptual Comparisons

Here, we clarify the conceptual motivation behind NOOCH, in particular as it relates to two (excellent) benchmarks aiming to do realistic robustness evaluation: WILDS [2] and Imagenet-A [3]. First, we note that NOOCH’s focus on OOC examples/spurious correlations is unique among these datasets (WILDS focuses on distribution shift, and Imagenet-A focuses on all examples with high loss, which may or may not be OOC), and as such NOOCH provides a useful resource for evaluation among this burgeoning line of work.

Additionally, we believe that our work provides essential insight into formalizing and interpreting example-level failure modes, an important aspect of model evaluation. We contrast this with WILDS, which specifies distribution-level shifts in generative process: many of the errors caused by WILDS-like distribution-level shifts may be mediated by (NOOCH-like) example-level mechanisms. For instance, a distribution shift may cause a shift in some underlying object co-occurrences, which then causes individual errors on some examples (e.g. where some important context object is not present). We also contrast this with Imagenet-A, which collects the highest-loss examples without specifying any failure mode in particular: some of these errors may be explained by (NOOCH-like) example-level mechanisms. We hope that by proposing and analyzing example-level mechanisms for model failure, NOOCH provides a more interpretable and actionable criterion for improving model robustness. We will include this analysis and clarification of our motivation in the next draft of the paper.

## Annotations

We discuss the common reviewer concern that the methods proposed in this paper rely on auxiliary annotations. Reviewers argue that 1) obtaining these annotations may be expensive, and 2) relying on annotations limits the applicability of these methods.

With regard to 1), we note that it is not required to have annotations (e.g. segmentions, captions) on the entire training set; rather, only a reasonably-sized evaluation set with sufficient positive and negative examples requires annotation. In many cases, this may keep costs manageable.

With regard to 2), we note that while not every dataset is fully annotated, a number of datasets exist which are at least partially annotated with segmentation (e.g. CityScapes, ADE20K, Pascal-VOC, SUN RGB-D) or captioning data (SentiCAP, Conceptual Captions, WikiCaps, Flickr30K). Additionally, we hope that this work provides inspiration for others to develop their own methods for automatically identifying OOC examples using whatever form of auxiliary information is available in their application.

At a higher level, we believe that reliance on annotations is a feature, rather than a bug, of our work. It is possible to imagine methods that, for instance, rely on ML-generated annotations, rather than human generated ones - for instance, using an automatic captioning model to generate captions. However, this type of approach has a clear flaw: the OOC examples will be exactly the ones the captioning model struggles to handle correctly. This kicks the can down the road, where our evaluation of our model’s OOC performance depends first on evaluating the OOC performance of the captioning model; at some point, grounding this out in human evaluation seems unavoidable. While this may not be cheap, evaluating a model’s robustness in deployment situations is already an expensive task. It is expensive to (for instance) employ ML engineers to do quality-control testing of models before they are deployed; obtaining human annotations for an evaluation set such as NOOCH may be cheaper, and is certainly more systematic. In practice, evaluating OOC performance may never be fully automated, but we hope that the work in this paper provides a useful step towards a more scalable approach.

### Citations

[1] Sagawa et al. Distributionally Robust Neural Networks for Group Shifts: On the Importance of Regularization for Worst-Case Generalization.

[2] Koh et al. WILDS: A Benchmark of in-the-Wild Distribution Shifts.

[3] Hendrycks et al. Natural Adversarial Examples.

---

### Decision · Program_Chairs · 2021-09-27

**Decision:**

Accept (Poster)

**Comment:**

Meta-review: In the context of image classification, the authors propose a method for constructing challenge sets of natural out-of-context examples from bounding-box annotations. They apply the method to COCO to construct a suite of challenge test sets, and evaluate their new benchmark task against other algorithms. All reviewers agreed on technical soundness. Most reviewers (h5rh, LYwb, wXHX) agreed that a benchmark of natural out-of-context examples is a valuable contribution. Reviewer 25HS questions the motivation given that other natural challenge sets exist, but in their rebuttal the authors differentiate by focusing on context shifts specifically. Reviewers 25HS and h5rh argue that the method is too specific to COCO (e.g. due to reliance on bounding boxes), but after rebuttal, h5rh doesn't think this detracts significantly and I tend to agree: the methodology seems secondary to the result, i.e. a potentially-useful new benchmark. On balance, I recommend acceptance.